# Influence of the Season on the Nutritive Value and Gas Production of *Opuntia ficus-indica* and *Agave americana* L. in Ruminant Feed

**DOI:** 10.3390/ani13061008

**Published:** 2023-03-10

**Authors:** Cristiana S. A. M. Maduro Dias, Helder P. B. Nunes, Carlos F. M. Vouzela, João S. Madruga, Alfredo E. S. Borba

**Affiliations:** Institute of Agricultural and Environmental Research and Technology (IITAA), Faculty of Agricultural and Environmental Sciences, University of the Azores, 9700-042 Angra do Heroísmo, Portugal

**Keywords:** diet optimisation, alternative feed sources, animal nutrition, ruminant nutrition, sustainability, dry season

## Abstract

**Simple Summary:**

The *Opuntia ficus-indica*, and the *Agave americana* L., are invasive plants, widespread across various countries, that are used by farmers as an alternative water source for ruminants in the summer. The goal of this study is to assess their nutritive value and potential throughout the year, so that their use in diets can be optimized. Both plants are very rich in water, minerals, and energy, but low in crude protein and fibre. Although a decrease in digestibility is observed in the summer, both plants can be a viable feed supplement for ruminants in both the winter and the summer, providing a cheap and eco-sustainable alternative water source. *Opuntia ficus-indica*, and the *Agave americana* L., can be fed to ruminants when associated with dried fodder, or incorporated into high crude protein diets for ruminants.

**Abstract:**

Using invasive plants in animal production can provide an economical and eco-sustainable competitive advantage in a globalized market. The *Opuntia ficus-indica* and the *Agave americana* L. are invasive plants historically used by Azorean farmers as an alternative ruminant water source in the summer. This study aims to better understand their properties and how they vary throughout the year, so their use to complement animal diets can be optimised. Six samples of each species were collected on the Terceira Island during 2 growth seasons: winter (January 2021) and summer (September 2021), and their chemical composition, in vitro digestibility, and gas production, were determined. Significant (*p* < 0.05) differences were found in all parameters between the summer and the winter, with larger variations in both fibre (NDF and ADF) and digestibility parameters found between the *Opuntia* and the *Agave*. Gas production was greater in the summer for *Opuntia* and in the winter for *Agave*. Even though the digestibility was lower in the summer, we found that both plants offer a viable ruminant feed complement in both seasons, providing a cheap and eco-sustainable alternative water source, that can be associated to dry forage and compound feeding stuff with a high crude protein content when designing ruminant diets.

## 1. Introduction

Interest in plants whose metabolic pathway for the synthesis of carbohydrates is the crassulacean acid metabolism (CAM) has been increasing in the scientific community in recent times. The interest stems essentially from two reasons. The first is the elevated tolerance these plants have to high temperatures, UV-B radiation, and drought or irregular rain conditions, by virtue of their ability to quickly take up and store water in above ground succulent tissues, while tolerating large losses in water mass fraction [1]. Their phenological and physiological adaptations allow these species to be very water efficient and to thrive in arid and semi-arid regions [2], especially during dry spells [3], when the production of more succulent food plants is severely limited. The second reason to study CAM plants is their ability to adapt to the most diverse types of climatic environment, as illustrated by the fact that these plants are widely distributed in various countries across the world, either as indigenous, alien, wild, or domesticated species [4]. Given their resilience and adaptability as invasive plants, CAM plants are of great concern, as they are difficult to control and compete with endemic vegetation, adversely affecting the production and quality of forages [5,6]. However, from an animal production perspective, their high palatability, production potential, and considerable survival propagation capacity under harsh dry conditions can make them a valuable resource for farmers that have access to them, as an emergency forage to prevent the disastrous consequences of droughts [7]. Indeed, they can be used as a complementary water source and, if combined with a protein source, constitute a complete feed [8].

The islands of Macaronesia contain a variety of invasive plants, including CAM plants, notably the genera *Agave* (Agavaceae) and *Opuntia* (Cataceae) that, to a greater or lesser extent, endanger its natural forest. Furthermore, the region is, like the rest of the world, suffering the effects of climate change, with the islands facing increasing periods of drought. As a result, several authors have been studying ways of taking advantage of CAM plants to complement animal feeds, namely as fibre and water sources in an effort to both contain their spread and, simultaneously, gain economic and environmental benefit [9].

*Opuntia ficus-indica*, in particular, has been gradually gaining economic importance in the international scientific community, through FAO, which contributes to the diffusion of this cultivation [10]. Ever since its introduction in the Iberian Peninsula, *Opuntia ficus-indica*, commonly known in Portugal as “figueira da índia”, has taken a prominent position from botanical, economical, and agricultural perspectives [11]. In turn, the *Agave americana* L., commonly known in Portugal as ”agave”, “piteira”, or “babosa”, was introduced in the Azores as an ornamental plant, but quickly spread as an invasive species in some islands, namely Graciosa and, especially, Santa Maria, occupying vast ranges of fallow lands.

The goal of this study is to evaluate *Agave americana* L. and *Opuntia ficus-indica*, present in the three archipelagos of Maraconesia’s European portion (Canary Islands, Azores and Madeira), as a possible animal feed complement, analysing their chemical composition and how it varies over two seasons of the year: winter and summer.

## 2. Materials and Methods

### 2.1. Sample Collection and Preparation

This study was conducted at the Animal Nutrition Lab, Department of Agricultural Sciences, University of the Azores, located in Angra do Heroísmo, Terceira, Azores, Portugal. The Azores are an archipelago of volcanic origin, located in the North Atlantic, whose soils can be characterised as andosols, having originated from modern volcanic materials, and evolved under temperate and humid Atlantic climate conditions [12]. For this study, plant samples were harvested on the Terceira Island, the third island of the Azorean archipelago in terms of size (402 km^2^), during 2 growth seasons: winter (January 2021) and summer (September 2021). Six plants of each species were collected in each season. In the case of *Opuntia,* cladodes were collected on the middle of the plant, whereas, with *Agave*, the whole blade was used. The minimum, maximum, and average temperature (°C), as well as the accumulated precipitation (mm) were also collected for one year, between October 2020 to September 2021 (Figure 1), to provide additional context to the study. The climatic data were obtained by the Portuguese Institute for Sea and Atmosphere, I. P. (IPMA, IP) in Angra do Heroísmo.

### 2.2. Chemical Composition

The plant samples were dried in a forced-air oven at 65 °C until constant weight. They were then ground through a 1-mm screen using a Retsch mill. The chemical characterisation of the plants was carried out using the Weende system. The dry matter (DM, method 930.15), crude protein (CP, method 954.01), ether extract (EE, method 920.39), and total ash (method 942.05) were determined according to the standard methods of [13]. The determination of the neutral detergent fibre (NDF), acid detergent fibre (ADF), and acid detergent lignin (ADL) were carried out according to [14]. Both NDF and ADF are expressed without residual ash.

### 2.3. In Vitro Methods

The in vitro dry matter digestibility and organic matter digestibility was measured according to the method of [15], modified by [16].

Gas production was measured as described by [17]. Two hundred mg of sampled dry matter was weighed in triplicate and placed in a glass syringe to which 30 mL of a mixture of rumen liquor (Menke medium mixture) was added and kept in CO_2_. The inoculant medium was prepared using the buffer solutions, and the rumen fluid as described by Menke et al. [17], mixing rumen fluid with buffer solutions (reduced and mineral solutions) in a ratio of 1:2 *v*/*v*.

Subsequently, the glass syringe was incubated at 39 ± 0.5 °C in an electrically heated isothermal oven equipped with a rotor, which rolled continuously at 1–2 rpm. Gas production was measured at 4, 8, 12, 24, 48, 72, and 96 h after the onset of incubation. Differences in the composition and activity of the rumen liquor were controlled by nine parallel measurements, a blank test, and incubation of a roughage and a concentrate standard.

Data for gas production were fitted to the gas production kinetics curve of Ørskov and McDonald (1979) [18]: $y=a+b\left(1-e^{\left(-ct\right)}\right)$.

Where *y* is the gas production at time *t*; *a* is the gas production of the immediately soluble fraction (mL 200 mg^−1^ DM); *b* is the gas production of the insoluble fraction (mL 200 mg^−1^ DM); *c* is the gas production rate constant for the insoluble fraction (mL h^−1^), and Lag time is the incubation time (in hours).

The rumen fluid for each experiment (digestibility and gas production) was collected as described by Borba et al. [19] in the local slaughterhouse, where the following conditions were observed: for each experiment, rumen was collected from 5 healthy dairy cattle, the animals had been fed ryegrass (*Lolium multiflorum*) and silage-based corn. Rumen fluid was collected within 10 min of slaughter, filtered of cheesecloth, and preserved at 39 °C under anaerobic conditions, being delivered to the animal nutrition laboratory within 30 min after being collected [20].

### 2.4. Data Analysis

All data were tested for normality used Shapiro–Wilk test in order to comply with the ANOVA conditions. The nutritive value and digestibility data were subjected to analysis of variance using the following model: $Y_{i}=µ+Z_{i}+S_{j}+P_{k}+RE_{ijk}$, where *Y* is the dependent variable, *µ* is the seasonal variations of the different parameters of overall mean, *Z_i_* is the species effect, *S_j_* is the season effect, *P_k_* is the interaction between species and season, and *RE_ijk_* is the random residual error assumed to be normally and independently distributed. The Tukey test was used to determine mean differences at *p* ≤ 0.05. The gas production variation metrics were not normally distributed and could not be normalised by transformation to allow for parametric analysis. Consequently, non-parametric methods were used for gas production, namely Kruskal–Wallis one-way analysis of variance by ranks and multiple comparisons between the samples with Kruskal–Wallis. Mean differences were considered statistically significant when the *p*-value obtained was lower than the significance level, which was set to 0.05. All statistical analyses were performed by IBM SPSS Statistics v24 program (SPSS Inc., Chicago, IL, USA).

## 3. Results

The results of the chemical composition and dry and organic matter digestibility for the two species being studied, for both the winter and the summer, are present in Table 1 and Table 2, respectively.

The minimum DM content (7.12%) was found during the summer for *Opuntia ficus-indica*, while the maximum dry matter value was 11.70% for *Agave americana* L., during the winter period. The fibre content (NDF), as expected, reached the maximum values during the summer for both plants. The minimum value of ADF (12.46%) was found in the summer period, in *Opuntia ficus-indica*, whereas the maximum content for *Agave americana* L. was 24.71%, also in the summer. Significant differences (*p* < 0.05) between species and period were observed in the DM, ADF, and NDF. Out of the chemical parameters that were studied, the high levels of ash present in both plants, which varied between 10.97% and 17.00% for *Opuntia ficus-indica* and for *Agave americana*, respectively, stand out.

Both plants showed dry matter digestibility above 80% during the winter. During the summer, both %DMD and %OMD decreased compared to the winter period, with the minimum OMD value being 58.12% for *Opuntia ficus-indica*. Significant dry and organic matter digestibility differences (*p* < 0.05) were found between species and periods.

The accumulated values of gas production by each of the plants over 96 h are shown in Figure 2. The gas production curves of *Opuntia ficus-indica* have a very similar behavior in both periods. The volume of gas produced by *Agave americana* L. in winter is higher than that observed in the summer.

Out of the parameters evaluated in the kinetics of in vitro gas production (Table 3), no significant differences were observed between the lag time values. Except for parameter *b*, for *Opuntia ficus-indica*, all the other parameters differed (*p* < 0.05) statistically between the two seasons of the year under analysis.

After adjusting the curve to the gas production model proposed by Ørskov and McDonald [18] (Table 4), we observe that, except for the 4 and 72 h, there were no significant (*p* < 0.05) differences between the periods in the case of *Opuntia ficus-indica*. Regarding *Agave americana* L., the gas production varies significantly (*p* < 0.05) over the 96 h of gas production.

Different letters next to the respective value indicate significant differences in the nutritive parameters among sampling dates.

## 4. Discussion

Plants are made up of an insoluble fraction, the cell wall, which is resistant to ruminant digestion and has lower degradation potential, and a soluble fraction, the cell content, with high digestibility. Non-conventional plants typically present a higher percentage of cell wall compared to cell content than other forages, like ryegrass and clovers. Besides the species, other factors influence the plant quality, such as the state of maturity, the handling, and the quality of the soil, which impacts the leaf area and photosynthetic capacity of the plants [21]. An increase of the cellulose, hemicellulose, and lignin, typically occurs during the dry seasons, making the forage more resistant, due to an increase of the cellulose and lignin connections, which makes the processes of ingestion, rumination, and fermentation by microorganisms harder [22]. CAM plants can produce up to 5 times more DM per millimetre of rainfall when compared to other plants, with [23] noting that, besides being an excellent water source, *Opuntia* also features high dry matter digestibility coefficients and is a high energy source, which is corroborated by [24] in a study made in north-eastern Brazil. Likewise, [25] reports that *Agave* has high water content and is a high energy source, in a study made with different species of the plant. As a result of these features and their resilience, CAM plants have historically played an important role in regions of arid and semi-arid environments [26], being used as a supplement in animal feed [27,28,29], in the context of both subsistence crops and market-oriented agriculture, thus increasing the food security of populations in agriculturally marginalized areas [24].

The chemical composition between the two species being studied, in two different periods of the year, is compared in Table 1. In line with the findings of [30], who noted CAM plants are characterised by a high water and ash content and low CP and NDF, both plants were found to have low DM content, with *Opuntia* averaging 7.12% in the winter and 7.48% in the summer, which are similar to the values reported by [31], of DM values varying between 7 and 15%.

In turn, the DM variation registered in *Agave* between the two seasons was only 0.5%, being 11.65% in the summer and 11.70% in the winter. Looking at the climate data, it can be observed that the amount of precipitation during the month of December was below average, whereas the precipitation during the summer and especially in August was quite abundant and well above average, even surpassing that of December. The combination of high precipitation and ideal temperature for growth allowed for the plants to develop more than usual, minimizing differences between seasons regarding certain parameters like cell content. The crude protein content found in these CAM plants was extremely low, independent of the season, with the maximum value being 6.30% for *Agave* in the summer. All values were below 7%, which is usually considered the minimum required value for normal function of ruminal microorganisms [32,33]. In the case of *Opuntia*, [34] reported that varieties could have variable amounts of CP and found that some clones from Brazil had over 11% DM of CP.

As can also be seen in Table 1, both *Agave americana* and *Opuntia ficus-indica* presented low values of NDF, with the particularity of there being an increase between the winter and the summer. For *Agave*, the observed values were 25.56% DM in the winter and 32.14% DM in the summer. For *Opuntia*, the values were 19.75% DM in the winter and 33.12% DM in the summer. This difference is broader than that reported by [30,35], who observed values ranging between 23.88% DM and 31.4% DM.

Although we found that both plants have a low nutritional value, they have good palatability and a high-water content [36], making sense to use them as a complementary source of water [37] in periods of shortage of feed or lack of water for animals, since water is one of the main limiting factors in animal production [30].

Regarding dry and organic matter digestibility (DMD and OMD, respectively), the values were found to be high in both summer and winter (Table 2). The similarity can be explained by the high rainfall in August and in the previous spring, which allowed the CAM to store considerable amounts of water and, consequently, to have greater digestibility.

For *Agave*, the DMD varied between 81.30% DM in the winter and 71.36% DM in the summer, while the OMD varied between 79.85% DM in the winter and 68.99% DM in the summer, in the same range of results found by [38], in a study of nutritional potential of some invasive species of Macaronesia for ruminants. Both plants had statistically significant (*p* < 0.05) differences between the summer and the winter for NDF, ADF, DMD, and OMD (Table 1 and Table 2). *Opuntia* is highly digestible [30,38]. Its results of DMD varied between 83.13% in the winter and 67.55% DM in the summer and 71.5% DM and 58.12% DM for OMD, respectively. These values are consistent with those reported by [39], who obtained values of 65% DM for *Opuntia*.

The amount of gas produced in in vitro fermentation reflects the extent of fermentation and digestibility of a forage [39] and is directly linked to the rate at which the substrate is degraded [40]. According to [41], the fermentation of the soluble and quickly fermentable fraction of the substrate, i.e., the soluble carbohydrates, and the synthesis of the microbial protein, both happen during the initial stages of incubation. Once this stage is complete, the fermentation of the insoluble fraction with degradation potential, such as the NDF fraction, begins.

The in vitro gas production simulates the ruminal fermentation process and has been used to assess the greenhouse gas production potential of feeds. It is similar to the ruminal process, since the gas (CO_2_ and CH_4_) is produced by the carbohydrate fermentation. The results of the in vitro gas production for both *Opuntia* and *Agave* are shown in Table 3, where we can see that *Agave* has a lag time of 0h in the winter, which indicates that the fermentation starts as soon as the element is incubated. Some authors note the high saponin content in this plant [42,43], therefore it would be interesting to determine anti-nutritive substances in future works. The value of the *a* constant of the reaction kinetics in *Agave* varied between 3.63 mL/0.2 g DM in the winter and −1.47 mL/0.2 g DM in the summer, with statistically significant (*p* < 0.05) differences between the seasons. Conversely, there was no statistically significant (*p* < 0.05) difference in the case of *Opuntia*, with a value of −4.87 mL/0.2 g DM registered in the winter and −6.06 mL/0.2 g DM in the summer. A positive *a* indicates that the component started to degrade quickly, whereas a negative value means there was an initial stage without cell wall degradation, called *lag* phase [44]. The *Agave* had an elevated soluble fraction in the winter (3.63) in comparison to the other results, which is reflected in the initial gas production amount which was found to be over 20 mL/200 g DM.

For parameter *b*, both *Opuntia* and *Agave* showed significant (*p* < 0.05) differences between seasons, which is an indicator of different fermentation patterns [45].

The cumulative fitted gas production value, summarized in Table 4 and Figure 2, showed significant (*p* < 0.05) differences being between seasons for the *Agave*, with a noticeably higher gas production in the winter, whereas *Opuntia* had a higher production in the winter, at 4 h, 72 h, and 96 h. In the case of *Opuntia*, it was observed that 85% of the gas recorded after 72 h was produced in the first 24 h. This is consistent with the findings of [46,47,48], who state that cactus nutrients are degraded rapidly in the rumen between 6 h and 12 h, with little nutrient degradation after 24 h.

## 5. Conclusions

*Opuntia* and *Agave* are very rich in water, and can be used in animal feed, especially during periods of severe food shortages or in drought situations, where water is scarce. The introduction of these plants in animal feed implies that the animals have a protein supplementation, due to the low protein and fibre content of both plants.

## Figures and Tables

**Figure 1 animals-13-01008-f001:**
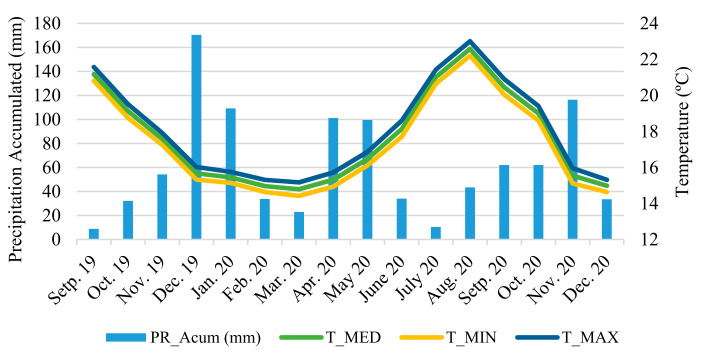
Climatic parameters for minimum, maximum and average temperature (°C), and accumulated precipitation (mm).

**Figure 2 animals-13-01008-f002:**
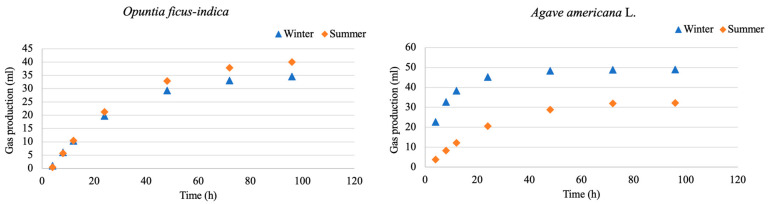
Cumulative fitted values of gas production of the different plants in the two periods.

**Table 1 animals-13-01008-t001:** Chemical Composition of the different plants to the different periods.

Parameter	*Opuntia ficus-indica*		*Agave americana* L.		SEM	*p*-Value		
	Period		Period			Species	Period	Species × Period
	Winter	Summer	Winter	Summer				
DM (%)	7.12 ^a^	7.48 ^a^	11.70 ^b^	11.65 ^b^	0.713	>0.05	>0.05	<0.05
CP (%DM)	4.15 ^a^	4.19 ^a^	5.16 ^a^	6.30 ^a^	0.224	>0.05	>0.05	>0.05
NDF (%DM)	19.75 ^a^	33.12 ^b^	25.56 ^c^	32.14 ^b^	1.41	<0.05	<0.05	<0.05
ADF (%DM)	14.80 ^a^	12.46 ^b^	22.51 ^c^	24.71^d^	1.74	<0.05	<0.05	<0.05
ADL (%DM)	3.47 ^a^	3.84 ^a^	4.68 ^a^	4.04 ^a^	0.544	>0.05	>0.05	>0.05
EE (%DM)	1.33 ^a^	1.11 ^a^	1.57 ^a^	1.73 ^a^	0.023	>0.05	>0.05	>0.05
Ash (%DM)	16.61 ^a^	17.00 ^a^	11.00 ^a^	10.97 ^a^	0.633	>0.05	>0.05	>0.05

DM: dry matter, CP: crude protein, NDF: neutral detergent insoluble fiber, ADF: acid detergent insoluble fiber, ADL: acid detergent lignin, EE: ether extract. SEM: standard error of the mean. Different letters next to the respective value indicate significant differences in the nutritive parameters among sampling dates. *p* < 0.05 significant differences were found.

**Table 2 animals-13-01008-t002:** Percentage dry matter digestibility (%DMD) and percentage organic matter digestibility (%OMD) of the different plants to the different periods.

Parameter	*Opuntia ficus-indica*		*Agave americana* L.		SEM	*p*-Value		
	Period		Period			Species	Period	Species × Period
	Winter	Summer	Winter	Summer				
DMD (%)	83.13 ^a^	67.55 ^b^	81.30 ^c^	71.36 ^d^	1.17	<0.05	<0.05	<0.05
OMD (%)	71.5 ^a^	58.12 ^b^	79.85 ^c^	68.99 ^d^	1.89	<0.05	<0.05	<0.05

DMD: dry matter digestibility, OMD: organic matter digestibility. SEM: standard error of the mean. Different letters next to the respective value indicate significant differences in the nutritive parameters among sampling dates. *p* < 0.05 significant differences were found.

**Table 3 animals-13-01008-t003:** In vitro gas production kinetics parameters of the different plants in the two periods.

Parameter	*Opuntia ficus-indica*		*Agave americana* L.	
	Period		Period	
	Winter	Summer	Winter	Summer
*a* (mL/0.2 g DM)	−4.87 ^a^	−6.06 ^a^	3.63 ^a^	−1.47 ^b^
*b* (mL/0.2 g DM)	40.64 ^a^	47.70 ^b^	45.38 ^a^	35.40 ^b^
*c* (mL/h)	0.0389 ^a^	0.0355 ^a^	0.175 ^a^	0.041 ^b^
Lag Time (h)	3.18 ^a^	3.82 ^b^	0 ^a^	1 ^a^
RSD	1.56	2.23	1.40	1.11

*a*: gas production of the immediately soluble fraction (mL/0.2g DM), *b*: gas production of the insoluble fraction (mL/0.2 g DM), *c*: gas production rate constant for the insoluble fraction (mL/h); Lag *t*: time it takes to produce gas (h); RSD = residual standard deviation. Different letters next to the respective value indicate significant differences in the nutritive parameters among sampling dates.

**Table 4 animals-13-01008-t004:** Cumulative fitted values of gas production of the different plants in the two periods.

Incubation Time (h)	*Opuntia ficus-indica*		*Agave americana* L.	
	Period		Period	
	Winter	Summer	Winter	Summer
4	1.024 ^a^	0.398 ^b^	22.680 ^a^	3.830 ^b^
8	6.056 ^a^	5.698 ^a^	32.675 ^a^	8.340 ^b^
12	10.340 ^a^	10.442 ^a^	38.290 ^a^	12.170 ^b^
24	19.762 ^a^	21.228 ^a^	45.290 ^a^	20.550 ^b^
48	29.286 ^a^	32.892 ^a^	48.415 ^a^	28.880 ^b^
72	33.050 ^a^	37.882 ^b^	48.905 ^a^	32.020 ^b^
96	34.580 ^a^	40.022 ^b^	48.990 ^a^	32.210 ^b^

Different letters next to the respective value indicate significant differences in the nutritive parameters among sampling dates.

## Data Availability

The data presented in this study are available on request from the corresponding author.

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
