# Peer review of "Influence of the Season on the Nutritive Value and Gas Production of Opuntia ficus-indica and Agave americana L. in Ruminant Feed"

_animals, 2023, doi:10.3390/ani13061008_

Round 1

Reviewer 1 Report

The article is adequate and reasoned, with a hypothesis, objective methodology, results and clear conclusions. Perhaps, what could be adjusted are the NDF and ADF data for ashes and proteins in order to have a more realistic determination of the chemical composition and digestibility. However, this alteration would not change, in my opinion, the final result even if in trend.

Author Response

Thank you for your feedback.

Reviewer 2 Report

This work investigates the nutritional value and digestibility of Opuntia ficus-indica, and Agave americana L., collected in summer and winter in the Azores Islands. Results showed that these succulent forage plants possess low crude protein but acceptable levels of fiber and very high ash.

The manuscript presents some problems in different sections. The title does align well with the objectives and the works carried out. The aims, key data, and conclusions of the manuscript are not well described in the abstract; therefore, it is hard to understand the general idea and conclusions by reading this section. Authors must indicate in greater detail what was done? What data were collected, how were they analyzed? What are the results and the implications? Therefore, the abstract needs to be rewritten with better clarity.

The article presents its introduction in a shallow manner with weak justification that motivated the study. Also, the introduction lacks flow and was written very generally. It should focus more on previous studies and should be emphasized that Opuntia and Agave are important alternative for farmers with access to these forages due to its high production potential and considerable survival and propagation capacity under conditions of scarce rain and high temperatures. Also, it is convenient to describe the amount of these plants used in livestock diets in previous works. Additionally, the authors must indicate in the introduction the high palatability of these succulent forages and the low DM, NDF, and CP contents of Opuntia and Agave which are insufficient for adequate animal performance.

In my opinion, one major shortcoming of this paper is the statistical analysis. Authors must include in the statistical model the effect of species, season and the forage x season interaction. Therefore, in tables, authors are encouraged to add three more columns to indicate the effect of forage species, season and species x season interaction.

An additional serious shortcoming is the limited information regarding the sampling process. Please indicate the following:

How many plants were sampled per season? What cladodes were collected (top or middle of the plant. In the case of Agave, what part of the blade was used for sampling collection.

The methodology is presented ambiguously; authors must indicate the following:

 -Indicate what procedure was used to test normality of data.

-Define nutrients in the bottom of Table 1

-Indicate type of soil in the study site.

Indicate how many grams of sample were used for in vitro gas production.

-Indicate number of replication for the in vitro gas production determination.

- Describe the anaerobic conditions during the incubation.

-Indicate the anaerobic buffer added to the rumen fluid.

Indicate hoe was the rumen fluid collected and transported to the laboratory

-Indicate if rumen liquor was from fasting bovines fed a standard diet.

-Indicate how ruminal liquor was filtered.

-Indicate if blanks were run to correct for the disappearance of organic matter (OM) and the production of gas and end-products.

-Indicate how the fermentation was stopped at 96 h

The discussion part needs much improvement from other publications using Opuntia or Agave forage plants in diets for cattle. Authors must discuss the vital role of these plants to the sustainability of farming systems in semi-arid regions, primarily as an energy source. Information about diets containing Opuntia and Agave plants with bulky and nitrogen sources.

Author Response

Thank you for your feedback. Please find the replies to each individual point in the attached file.

Round 2

Reviewer 2 Report

Authors have appropriately addressed all my observations. Few minor grammatical changes are needed.
